# A Unifying Framework for Spectrum-Preserving Graph Sparsification and Coarsening

**Gecia Bravo-Hermsdorff***
Princeton Neuroscience Institute
Princeton University
Princeton, NJ, 08544, USA
`geciah@princeton.edu`

**Lee M. Gunderson***
Department of Astrophysical Sciences
Princeton University
Princeton, NJ, 08544, USA
`leeg@princeton.edu`

## Abstract

How might one "reduce" a graph? That is, generate a smaller graph that preserves the global structure at the expense of discarding local details? There has been extensive work on both graph sparsification (removing edges) and graph coarsening (merging nodes, often by edge contraction); however, these operations are currently treated separately. Interestingly, for a planar graph, edge deletion corresponds to edge contraction in its planar dual (and more generally, for a graphical matroid and its dual). Moreover, with respect to the dynamics induced by the graph Laplacian (e.g., diffusion), deletion and contraction are physical manifestations of two reciprocal limits: edge weights of $0$ and $\infty$, respectively. In this work, we provide a unifying framework that captures both of these operations, allowing one to simultaneously sparsify and coarsen a graph while preserving its large-scale structure. The limit of infinite edge weight is rarely considered, as many classical notions of graph similarity diverge. However, its algebraic, geometric, and physical interpretations are reflected in the Laplacian pseudoinverse $\boldsymbol{L}^{\dagger}$, which remains finite in this limit. Motivated by this insight, we provide a probabilistic algorithm that reduces graphs while preserving $\boldsymbol{L}^{\dagger}$, using an unbiased procedure that minimizes its variance. We compare our algorithm with several existing sparsification and coarsening algorithms using real-world datasets, and demonstrate that it more accurately preserves the large-scale structure.

## 1 Motivation

Many complex structures and phenomena are naturally described as graphs (eg:[1] brains, social networks, the internet, etc). Indeed, graph-structured data are becoming increasingly relevant to the field of machine learning [2, 3, 4]. These graphs are frequently massive, easily surpassing our working memory, and often the computer's relevant cache [5]. It is therefore essential to obtain smaller approximate graphs to allow for more efficient computation.

Graphs are defined by a set of nodes $V$ and a set of edges $E \subseteq V \times V$ between them, and are often represented as an adjacency matrix $\boldsymbol{A}$ with size $|V| \times |V|$ and density $\propto |E|$. Reducing either of these quantities is advantageous: graph "coarsening" focuses on the former, aggregating nodes while respecting the overall structure, and graph "sparsification" on the latter, preferentially retaining the important edges.

Spectral graph sparsification has revolutionized the field of numerical linear algebra and is used, eg, in algorithms for solving linear systems with symmetric diagonally dominant matrices in nearly-linear time [6, 7] (in contrast to the fastest known algorithm for solving general linear systems, taking $\mathcal{O}(n^\omega)$-time, where $\omega \approx 2.373$ is the matrix multiplication exponent [8]).

Graph coarsening appears in many computer science and machine learning applications, eg: as primitives for graph partitioning [9] and visualization algorithms[2] [10]; as layers in graph convolution networks [3, 11]; for dimensionality reduction and hierarchical representation of graph-structured data [12, 13]; and to speed up regularized least square problems on graphs [14], which arise in a variety of problems such as ranking [15] and distributed synchronization of clocks [16].

A variety of algorithms, with different objectives, have been proposed for both sparsification and coarsening. However, a frequently recurring theme is to consider the graph Laplacian $L = D - A$, where $D$ is the diagonal matrix of node degrees. Indeed, it appears in a wide range of applications, eg: its spectral properties can be leveraged for graph clustering [17]; it can be used to efficiently solve min-cut/max-flow problems [18]; and for undirected, positively weighted graphs (the focus of this paper), it induces a natural quadratic form, which can be used, eg, to smoothly interpolate functions over the nodes [19].

Work on spectral graph sparsification focuses on preserving the Laplacian quadratic form $\vec{x}^\top L \vec{x}$, a popular measure of spectral similarity suggested by Spielman & Teng [6]. A key result in this field is that any dense graph can be sparsified to $\mathcal{O}(|V| \log |V|)$ edges in nearly linear time using a simple probabilistic algorithm [20]: start with an empty graph, include edges from the original graph with probability proportional to their effective resistance, and appropriately reweight those edges so as to preserve $\vec{x}^\top L \vec{x}$ within a reasonable factor.

In contrast to the firm theoretical footing of spectral sparsification, work on graph coarsening has not reached a similar maturity; while a variety of spectral coarsening schemes have been recently proposed, algorithms frequently rely on heuristics, and there is arguably no consensus. Eg: Jin & Jaja [21] use $k$ eigenvectors of the Laplacian as feature vectors to perform $k$-means clustering of the nodes; Purohit et al. [22] aim to minimize the change in the largest eigenvalue of the adjacency matrix; and Loukas & Vandergheynst [23] focuses on a "restricted" Laplacian quadratic form.

Although recent work has combined sparsification and coarsening [24], they used separate algorithmic primitives, essentially analyzing the serial composition of the above algorithms. The primary contribution of this work is to provide a unifying probabilistic framework that allows one to simultaneously sparsify and coarsen a graph while preserving its global structure by using a *single* cost function that preserves the Laplacian pseudoinverse $L^\dagger$.

Corollary contributions include: **1)** Identifying the limit of infinite edge weight with edge contraction, highlighting how its algebraic, geometric, and physical interpretations are reflected in $L^\dagger$, which remains finite in this limit (Section 2); **2)** Offering a way to quantitatively compare the effects of edge deletion and edge contraction (Section 2 and 3); **3)** Providing a probabilistic algorithm that reduces graphs while preserving $L^\dagger$, using an unbiased procedure that minimizes its variance (Sections 3 and 4); **4)** Proposing a more sensitive measure of spectral similarity of graphs, inspired by the Poincaré half-plane model of hyperbolic space (Section 5.3); and **5)** Comparing our algorithm with several existing sparsification and coarsening algorithms using synthetic and real-world datasets, demonstrating that it more accurately preserves the large-scale structure (Section 5).

## 2 Why the Laplacian pseudoinverse

Many computations over graphs involve solving $L\vec{x} = \vec{b}$ for $\vec{x}$ [25]. Thus, the algebraically relevant operator is arguably the Laplacian pseudoinverse $L^\dagger$. In fact, its connection with random walks has been used to derive useful measures of distances on graphs, such as the well-known effective resistance [26], and the recently proposed resistance perturbation distance [27]. Moreover, taking the pseudoinverse of $L$ leaves its eigenvectors unchanged, but inverts the nontrivial eigenvalues. Thus, as the largest eigenpairs of $L^\dagger$ are associated with global structure, preserving its action will preferentially maintain the overall "shape" of the graph (see Appendix Section G for details). For instance, the Fielder vector [17] (associated with the "algebraic connectivity" of a graph) will be

youtube.com/playlist?list=PLmfiQcz2q6d3sZutLri4ZAIDLqM_4K1p-

preferentially preserved. We now discuss in further detail why $\boldsymbol{L}^{\dagger}$ is well-suited for both graph sparsification and coarsening.

Attention is often restricted to undirected, positively weighted graphs [28]. These graphs have many convenient properties, eg, their Laplacians are positive semidefinite ($\vec{x}^{\top}\boldsymbol{L}\vec{x} \geq 0$) and have a well-understood kernel and cokernel ($\boldsymbol{L}\vec{1} = \vec{1}^{\top}\boldsymbol{L} = \vec{0}$). The edge weights are defined as a mapping $W\colon E \to \mathbb{R}_{>0}$. When the weights represent connection strength, it is generally understood that $w_e \to 0$ is equivalent to removing edge $e$. However, the closure of the positive reals has a reciprocal limit, namely $w_e \to +\infty$.

This limit is rarely considered, as many classical notions of graph similarity diverge. This includes the standard notion of spectral similarity, where $\widetilde{G}$ is a $\sigma$-spectral approximation of $G$ if it preserves the Laplacian quadratic form $\vec{x}^{\top}\boldsymbol{L}_G\vec{x}$ to within a factor of $\sigma$ for all vectors $\vec{x} \in \mathbb{R}^{|V_G|}$ [6]. Clearly, this limit yields a graph that does not approximate the original for any choice of $\sigma$: any $\vec{x}$ with different values for the two nodes joined by the edge with infinite weight now yields an infinite quadratic form. This suggests considering only vectors that have the same value for these two nodes, essentially contracting them into a single "supernode". Algebraically, this interpretation is reflected in $\boldsymbol{L}^{\dagger}$, which remains finite in this limit: the pair of rows (and columns) corresponding to the contracted nodes become identical (see Appendix Section C).

Physically, consider the behavior of the heat equation $\partial_t\vec{x} + \boldsymbol{L}\vec{x} = \vec{0}$: as $w_e \to +\infty$, the values on the two nodes immediately equilibrate between themselves, and remain tethered for the rest of the evolution.[3] Geometrically, the reciprocal limits of $w_e \to 0$ and $w_e \to +\infty$ have dual interpretations: consider a planar graph and its planar dual; edge deletion in one graph corresponds to contraction in the other, and vice versa. This naturally extends to nonplanar graphs via their graphical matroids and their duals [29].

Finally, while the Laplacian operator is frequently considered in the graph sparsification and coarsening literature, its pseudoinverse also has many important applications in the field of machine learning [30], eg: online learning over graphs [31]; similarity prediction of network data [32]; determining important nodes [33]; providing a measure of network robustness to multiple failures [34]; extending principal component analysis to graphs [35]; and collaborative recommendation systems [36]. Hence, graph reduction algorithms that preserve $\boldsymbol{L}^{\dagger}$ would be useful to the machine learning community.

## 3 Our graph reduction framework

We now describe our framework for constructing probabilistic algorithms that generate a reduced graph $\widetilde{G}$ from an initial graph $G$, motivated by the following desiderata: **1)** Reduce the number of edges/nodes (Section 3.1); **2)** Preserve $\boldsymbol{L}^{\dagger}$ in expectation (Section 3.2); and **3)** Minimize the change in $\boldsymbol{L}^{\dagger}$ (Section 3.3).

We first define these goals more formally. Then, in Section 3.4, we combine these requirements to define our cost function and derive the optimal probabilistic action (ie, deletion, contraction, or reweight) to perform to an edge.

### 3.1 Reducing edges and nodes

Depending on the application, it might be more important to reduce the number of nodes (eg, coarsening a sparse network) or the number of edges (eg, sparsifying a dense network). Let $r$ be the number of prioritized items reduced during a particular iteration. When those items are nodes, then $r = 0$ for a deletion, and $r = 1$ for a contraction. When those items are edges, then $r = 1$ for a deletion, however $r > 1$ for a contraction is possible: if the contracted edge forms a triangle in the original graph, then the other two edges will become parallel in the reduced graph (see Figure SI 3 in Appendix Section C). With respect to the Laplacian, this is equivalent to a single edge with weight given by the sum of these now parallel edges. Thus, when edge reduction is prioritized, a contraction will have $r = 1 + \tau_e$, where $\tau_e$ is the number of triangles in the original graph $G$ in which the contracted edge $e$ participates.

Note that, even when node reduction is prioritized, the number of edges will also necessarily decrease. Conversely, when edge reduction is prioritized, contraction of an edge is also possible, thereby reducing the number of nodes as well. For the case of simultaneously sparsifying *and* coarsening a graph, we choose to prioritize edge reduction, although nodes could also be a sensible choice.

## 3.2 Preserving the Laplacian pseudoinverse

Consider perturbing the weight of a single edge $e = (v_1, v_2)$ by $\Delta w$. The change in the Laplacian is

$$\boldsymbol{L}_{\tilde{G}} - \boldsymbol{L}_G = \Delta w \vec{b}_e \vec{b}_e^\top, \tag{1}$$

where $\boldsymbol{L}_{\tilde{G}}$ and $\boldsymbol{L}_G$ are the perturbed and original Laplacians, respectively, and $\vec{b}_e$ is the (arbitrarily) signed incidence (column) vector associated with edge $e$, with entries

$$(b_e)_i = \begin{cases} +1 & i = v_1 \\ -1 & i = v_2 \\ 0 & \text{otherwise.} \end{cases} \tag{2}$$

The change in $\boldsymbol{L}^\dagger$ is given by the Woodbury matrix identity[4] [39]:

$$\boldsymbol{L}_{\tilde{G}}^\dagger - \boldsymbol{L}_G^\dagger = -\frac{\Delta w}{1 + \Delta w \vec{b}_e^\top \boldsymbol{L}_G^\dagger \vec{b}_e} \boldsymbol{L}_G^\dagger \vec{b}_e \vec{b}_e^\top \boldsymbol{L}_G^\dagger. \tag{3}$$

Note that this change can be expressed as a matrix that depends *only* on the choice of edge $e$, multiplied by a scalar term that depends (nonlinearly) on the change to its weight:

$$\boldsymbol{\Delta L}^\dagger = \underbrace{f\left(\frac{\Delta w}{w_e}, w_e \Omega_e\right)}_{\text{nonlinear scalar}} \times \underbrace{\boldsymbol{M}_e}_{\text{constant matrix}}, \tag{4}$$

where

$$f = -\frac{\frac{\Delta w}{w_e}}{1 + \frac{\Delta w}{w_e} w_e \Omega_e}, \tag{5}$$

$$\boldsymbol{M}_e = w_e \boldsymbol{L}_G^\dagger \vec{b}_e \vec{b}_e^\top \boldsymbol{L}_G^\dagger, \tag{6}$$

$$\Omega_e = \vec{b}_e^\top \boldsymbol{L}_G^\dagger \vec{b}_e. \tag{7}$$

Hence, if the probabilistic reweight of this edge is chosen such that $\mathbb{E}[f] = 0$, then we have $\mathbb{E}[\boldsymbol{L}_{\tilde{G}}^\dagger] = \boldsymbol{L}_G^\dagger$, as desired. Importantly, $f$ remains finite in the following relevant limits:

$$\begin{array}{lll} \text{deletion:} & \frac{\Delta w}{w_e} \to -1, & f \to (1 - w_e \Omega_e)^{-1} \\ \text{contraction:} & \frac{\Delta w}{w_e} \to +\infty, & f \to -(w_e \Omega_e)^{-1}. \end{array} \tag{8}$$

Note that $f$ diverges when considering deletion of an edge with $w_e \Omega_e = 1$ (ie, an edge cut). Indeed, such an action would disconnect the graph and invalidate the use of equation 3 (see footnote 4). However, this possibility is precluded by the requirement that $\mathbb{E}[f] = 0$.

## 3.3 Minimizing the error

Minimizing the magnitude of $\boldsymbol{\Delta L}^\dagger$ requires a choice of matrix norm, which we take to be the sum of the squares of its entries (ie, the square of the Frobenius norm). Our motivation is twofold. First, the algebraically convenient fact that the Frobenius norm of a rank one matrix has a simple form, viz,

$$m_e \equiv \|\boldsymbol{M}_e\|_{\mathrm{F}} = w_e \vec{b}_e^\top \boldsymbol{L}_G^\dagger \boldsymbol{L}_G^\dagger \vec{b}_e. \tag{9}$$

Second, the square of this norm behaves as a variance; to the extent that the $\boldsymbol{M}_e$ associated to different edges can be treated as (entrywise) uncorrelated one can decompose multiple perturbations as follows:

$$\mathbb{E}\left[\left\|\sum \boldsymbol{\Delta L}^\dagger\right\|_{\mathrm{F}}^2\right] \approx \sum \mathbb{E}\left[\|\boldsymbol{\Delta L}^\dagger\|_{\mathrm{F}}^2\right], \tag{10}$$

which allows the single-edge results from Section 3.4 to be iteratively applied to our reduction algorithm, which has multiple reductions (Section 4). In Appendix Section A, we empirically validate this approximation using synthetic and real-world networks, showing that this approximation is either nearly exact or a conservative estimate.

For subtleties associated with edge contraction (see Appendix Section F, in particular equation 39).

## 3.4 A cost function for spectral graph reduction

Combining the discussed desiderata, we choose to minimize the following cost function:

$$\mathcal{C} = \mathbb{E}\left[\left\|\boldsymbol{\Delta L}^{\dagger}\right\|_{\mathrm{F}}^2\right] - \beta^2 \mathbb{E}[r], \tag{11}$$

subject to

$$\mathbb{E}\left[\boldsymbol{\Delta L}^{\dagger}\right] = \boldsymbol{0}, \tag{12}$$

where the parameter $\beta$ controls the tradeoff between number of prioritized items reduced $r$ and error incurred in $\boldsymbol{L}^{\dagger}$. This cost function naturally arises when minimizing the expected squared error for a given expected amount of reduction (or equivalently maximizing the expected number of reductions for a given expected squared error).

We desire to minimize this cost function over all possible reduced graphs. As, when reducing multiple edges, $\mathbb{E}[r]$ is additive and the expected squared error is empirically additive, we are able to decompose this objective into a sequence of minimizations applied to individual edges. Thus, minimization of this cost function for each edge acted upon can be seen as a probabilistic greedy algorithm for minimizing the cost function for the final reduced graph.

Here, we describe the analytic solution for the optimal action (ie, probabilistically choosing to delete, contract, or reweight) to be applied to a single edge. We provide the solution in Figure 1, and a detailed derivation in Appendix Section B.

For a given edge $e$, the values of $m_e$, $w_e\Omega_e$, and $\tau_e$ are fixed, and minimizing the cost function (11) (given (12)) results in a piecewise solution with three regimes, depending on the value of $\beta$: **1)** When $\beta < \beta_1(m_e, w_e\Omega_e, \tau_e) = \min(\beta_{1d}, \beta_{1c})$, $\beta$ is small compared with the error that would be incurred by acting on this edge, thus it should not be changed; **2)** When $\beta > \beta_2(m_e, w_e\Omega_e, \tau_e)$, $\beta$ is large for this edge, and the optimal solution is to probabilistically delete or contract this edge ($p_d + p_c = 1$; no reweight is required); and **3)** In the intermediate case ($\beta_1 < \beta < \beta_2$), there are two possibilities, depending on the edge and the choice of prioritized items: if $\beta_{1d} < \beta_{1c}$, the edge is either deleted or reweighted, and if $\beta_{1c} < \beta_{1d}$, the edge is either contracted or reweighted.

| $\beta < \beta_1$ | $p_d = 0, \quad p_c = 0, \quad \frac{\Delta w}{w_e} = 0$ |
|---|---|
| $\beta_{1d} < \beta_{1c}$ | $p_d = 1 - \frac{m_e}{(1-w_e\Omega_e)\beta}, \quad p_c = 0,$  $\frac{\Delta w}{w_e} = \left(1 - \frac{p_d}{1-w_e\Omega_e}\right)^{-1} - 1$ |
| $\beta_{1c} < \beta_{1d}$ | $p_d = 0, \quad p_c = 1 - \frac{m_e}{w_e\Omega_e\beta\sqrt{1+\tau_e}},$  $\frac{\Delta w}{w_e} = -\frac{p_c}{w_e\Omega_e}$ |
| $\beta > \beta_2$ | $p_d = 1 - w_e\Omega_e, \quad p_c = w_e\Omega_e$ |

(left column spanning brace: $\beta_1 < \beta < \beta_2$)

|  | prioritizing edges | prioritizing nodes |
|---|---|---|
| $\beta_{1d}$ | $\frac{m_e}{1-w_e\Omega_e}$ | $\infty$ |
| $\beta_{1c}$ | $\frac{m_e}{w_e\Omega_e}\frac{1}{\sqrt{1+\tau_e}}$ | $\frac{m_e}{w_e\Omega_e}$ |
| $\beta_2$ | $\frac{m_e}{w_e\Omega_e(1-w_e\Omega_e)}\frac{1}{1+\sqrt{1+\tau_e}}$ | $\frac{m_e}{w_e\Omega_e(1-w_e\Omega_e)}$ |

Figure 1: *Left*: **Minimizing $\mathcal{C}$ for a single edge** $e$. There are three regimes for the solution, depending on the value of $\beta$. When node reduction is prioritized, set $\tau_e = 0$. *Right*: **Values of $\beta$ dividing the three regimes**. Note that when edge reduction is prioritized, the number of triangles enters the expressions, and when node reduction is prioritized, there is no deletion in the intermediate regime. However, for either choice, both deletion and contraction can have finite probability, and the algorithm does not exclusively reduce one or the other. Thus, when simultaneously sparsifying and coarsening a graph, the prioritized items may be chosen to be either edges or nodes. We remark that the values of $\beta_{1d}$, $\beta_{1c}$, and $\beta_2$ might be of independent interest as measures of edge importance for analyzing connections in real-world networks.

## 3.5 Node-weighted Laplacian

When nodes are merged, one often represents the connectivity of the resulting graph $\widetilde{G}$ by a matrix of smaller size. To properly compare the spectral properties of $\widetilde{G}$ with those of the original graph $G$, one must keep track of the number of original nodes that comprise these "supernodes" and assign them proportional weights. The appropriate reduced Laplacian $\boldsymbol{L}_{\widetilde{G}}$ (of size $|V_{\widetilde{G}}| \times |V_{\widetilde{G}}|$) is then $\boldsymbol{W}_n^{-1} \boldsymbol{B}^\top \boldsymbol{W}_e \boldsymbol{B}$, where the $\boldsymbol{W}$ are the diagonal matrices of the node weights[5] and the edge weights of $\widetilde{G}$, respectively, and $\boldsymbol{B}$ is its signed incidence matrix with columns given by (2).

Moreover, one must be careful to choose the appropriate pseudoinverse of $\boldsymbol{L}_{\widetilde{G}}$, which is given by

$$\boldsymbol{L}_{\widetilde{G}}^\dagger = \left(\boldsymbol{L}_{\widetilde{G}} + \boldsymbol{J}\right)^{-1} - \boldsymbol{J}, \tag{13}$$

$$\boldsymbol{J} = \frac{1}{\vec{1}^\top \vec{w}_n} \vec{1}\vec{w}_n^\top, \tag{14}$$

where $\vec{w}_n \in \mathbb{R}_{>0}^{|V_{\widetilde{G}}|}$ is the vector of node weights. Note that $\boldsymbol{L}_{\widetilde{G}}^\dagger \boldsymbol{L}_{\widetilde{G}} = \boldsymbol{L}_{\widetilde{G}} \boldsymbol{L}_{\widetilde{G}}^\dagger = \boldsymbol{I} - \boldsymbol{J}$, the appropriate node-weighted projection matrix.

To compare the action of the original and reduced Laplacians on a vector $\vec{x} \in \mathbb{R}^{|V_G|}$ over the nodes of the original graph, one must "lift" $\boldsymbol{L}_{\widetilde{G}}$ to operate on the same space as $\boldsymbol{L}_G$. We thus define the mapping from original to coarsened nodes as a $|V_{\widetilde{G}}| \times |V_G|$ matrix $\boldsymbol{C}$, with entries

$$c_{ij} = \begin{cases} 1 & \text{node } j \text{ in supernode } i \\ 0 & \text{otherwise.} \end{cases} \tag{15}$$

The appropriate lifted Laplacian is $\boldsymbol{L}_{\widetilde{G},l} = \boldsymbol{C}^\top \boldsymbol{L}_{\widetilde{G}} \boldsymbol{W}_n^{-1} \boldsymbol{C}$. Likewise, the lifted Laplacian pseudoinverse is $\boldsymbol{L}_{\widetilde{G},l}^\dagger = \boldsymbol{C}^\top \boldsymbol{L}_{\widetilde{G}}^\dagger \boldsymbol{W}_n^{-1} \boldsymbol{C}$ (see Appendix Section C for a detailed rationale of these definitions).

# 4 Our graph reduction algorithm

Using this framework, we now describe our graph reduction algorithm. Similar to many graph coarsening methods [41, 42], we obtain the reduced graph by acting on the initial graph (as opposed to adding edges to an empty graph, as is frequently done in sparsification [43, 44]).

Care must be taken, however, as simultaneous deletions/contractions may result in undesirable behavior. Eg, while any edge that is itself a cut-set will never be deleted (as $w_e \Omega_e = 1$), a collection of edges that together make a cut-set might all have finite deletion probability. Hence, if multiple edges are simultaneously deleted, the graph could become disconnected. In addition, the single-edge analysis could underestimate the change in $\boldsymbol{L}^\dagger$ associated with simultaneous contractions. Eg, consider two highly-connected nodes that are each the center of a different community, and a third auxiliary node that happens to be connected to both: contracting the auxiliary node into either of the other two would be sensible, but performing both contractions would merge the two communities.

Algorithm 1 describes our graph reduction scheme. Its inputs are: $G$, the original graph; $q$, the fraction of sampled edges to act upon per iteration; $d$, the minimum expected decrease in prioritized items per edge acted upon; and `StopCriterion`, a user-defined function. With these inputs, we implicitly select $\beta$. Let $\beta_{\star,e}$ be the minimum $\beta$ such that $\mathbb{E}[r] \geq d$ for edge $e$. For each iteration, we compute $\beta_{\star,e}$ for all sampled edges, and choose a $\beta$ such that a fraction $q$ of them have $\beta_{\star,e} < \beta$. We then apply the corresponding probabilistic actions to these edges. The appropriate choice of `StopCriterion` depends on the application. Eg, if one desires to bound the accuracy of an algorithm that uses graph reduction as a primitive, limiting the Frobenius error in $\boldsymbol{L}^\dagger$ is a sensible choice (it is trivial to keep a running total of the estimated error, see Appendix Section A). On the other hand, if one would like the reduced graph to be no larger than a certain size, then one can simply continue reducing until this point. While both criteria may also be implicitly implemented via an upper bound on $\beta$, the relationship is nontrivial and depends on the structure of the graph.

The aforementioned problems associated with simultaneous deletions/contractions can be eliminated by taking a conservative approach: acting on only a single edge per iteration. However, this results in an algorithm that does not scale favorably for large graphs. A more scalable solution involves

**Algorithm 1** ReduceGraph
---
1: **Inputs:** graph $G$, fraction of sampled edges to act upon $q$, minimum $\mathbb{E}[r]$ per edge acted upon $d$, and a StopCriterion
2: Initialize $\widetilde{G}_0 \leftarrow G$, $t \leftarrow 0$, $stop \leftarrow$ False
3: **while not** (*stop*) **do**
4:     Sample an independent edge set
5:     **for** (edge $e$) **in** (sampled edges) **do**
6:         Compute $\Omega_e$, $m_e$ (see equations (7) and (9))
7:         Evaluate $\beta_{\star e}$, according to $d$ (see Tables in Figure 1)
8:     **end for**
9:     Choose $\beta$ such that a fraction $q$ of the sampled edges (those with the lowest $\beta_{\star e}$) are acted upon
10:     Probabilistically choose to reweight, delete, or contract these edges
11:     Perform reweights and deletions to $\widetilde{G}_t$
12:     Perform contractions to $\widetilde{G}_t$
13:     $\widetilde{G}_{t+1} \leftarrow \widetilde{G}_t$, $t \leftarrow t + 1$
14:     $stop \leftarrow$ StopCriterion($\widetilde{G}_t$)
15: **end while**
16: **return** reduced graph $\widetilde{G}_t$
---

carefully sampling the candidate set of edges. In particular, we are able to significantly ameliorate these issues by sampling the candidate edges such that they do not have any nodes in common (ie, the sampled edges form an independent edge set). Not only does this eliminate the possibility of "accidental" contractions, but, empirically, it also suppresses the occurrence of graph disconnections (the small fraction that become disconnected are restarted). At each iteration, our algorithm finds a random maximal independent edge set in $\mathcal{O}(|V|)$ time using a simple greedy algorithm.[6] In practice, the size of such a set scales as $\mathcal{O}(|V|)$ (although it is easy to find families for which this scaling does not hold, eg, star graphs). Our algorithm then computes the $\Omega_e$ and $m_e$ of these sampled edges, and acts on the fraction $q$ with the lowest $\beta_{\star e}$.

The main computational bottleneck of our algorithm is computing $\Omega_e$ and $m_e$ (equation (9)). However, we can draw on the work of [20], which describes a method for efficiently computing $\varepsilon$-approximate values of $\Omega_e$ for all edges, requiring $\widetilde{\mathcal{O}}(|E| \log |V|/\epsilon^2)$ time. With minimal changes, this procedure can also be used to compute approximate values of $m_e$ with similar efficiency (in Appendix Section F, we discuss the details of how to efficiently compute approximations of $m_e$). As we must compute these quantities for each iteration, we multiply the running time by the expected number of iterations, $\mathcal{O}(|E|/qd|V|)$. Empirically, we find that one is able to set $q \sim 1/16$ and $d \sim 1/4$ with minimal loss in reduction quality (see Appendix Section E). Thus, we expect that our algorithm could have a running time of $\widetilde{\mathcal{O}}(\langle k \rangle |E|)$, where $\langle k \rangle$ is the average degree. However, in the following results, we have used a naive implementation: computing $\boldsymbol{L}^\dagger$ at the onset, and updating it using the Woodbury matrix identity.

## 5 Experimental results

In this section, we empirically validate our framework and compare it with existing algorithms. We consider two cases of our general framework, namely graph sparsification (excluding regimes involving edge contraction), and graph coarsening (prioritizing reduction of nodes). In addition, as graph reduction is often used in graph visualization, we generated videos of our algorithm simultaneously sparsifying and coarsening several real-world datasets (see footnote 2 and Appendix Section I).

### 5.1 Hyperbolic interlude

When comparing a graph $G$ with its reduced approximation $\widetilde{G}$, it is natural to consider how relevant linear operators treat the same input vector. If the vector $\boldsymbol{L}_{\widetilde{G},l}\vec{x}$ is aligned with $\boldsymbol{L}_G\vec{x}$, the fractional error in the quadratic form $\vec{x}^\top \boldsymbol{L}\vec{x}$ is a natural quantity to consider, as it corresponds to the relative change in the magnitude of these vectors. However, it is not so clear how to compare output vectors that have

an angular difference. Here, we describe a natural extension of this notion of fractional error, which draws intuition from the Poincaré half-plane model of hyperbolic geometry. In particular, we choose the boundary of the half-plane to be perpendicular to $\vec{x}$ and compute the geodesic distance between $\boldsymbol{L}_G\vec{x}$ and $\boldsymbol{L}_{\widetilde{G},l}\vec{x}$, viz,

$$d_{\vec{x}}(\boldsymbol{L}_0, \boldsymbol{L}_1) \stackrel{\text{def}}{=} \operatorname{arccosh}\left(1 + \frac{\left\|(\boldsymbol{L}_0 - \boldsymbol{L}_1)\vec{x}\right\|_2^2 \left\|\vec{x}\right\|_2^2}{2\left(\vec{x}^{\intercal}\boldsymbol{L}_0\vec{x}\right)\left(\vec{x}^{\intercal}\boldsymbol{L}_1\vec{x}\right)}\right), \tag{16}$$

where $\boldsymbol{L}_0$ and $\boldsymbol{L}_1$ are positive definite matrices (for now).

We define the hyperbolic distance between these matrices as

$$d_h(\boldsymbol{L}_0, \boldsymbol{L}_1) \stackrel{\text{def}}{=} \sup_{\vec{x}} d_{\vec{x}}(\boldsymbol{L}_0, \boldsymbol{L}_1). \tag{17}$$

This dimensionless quantity inherits the following standard desirable features of a distance: symmetry and non-negativity, $d_h(\boldsymbol{L}_0, \boldsymbol{L}_1) = d_h(\boldsymbol{L}_1, \boldsymbol{L}_0) \geq 0$; identity of indiscernibles, $d_h(\boldsymbol{L}_0, \boldsymbol{L}_1) = 0 \iff \boldsymbol{L}_0 = \boldsymbol{L}_1$; and subadditivity, $d_h(\boldsymbol{L}_0, \boldsymbol{L}_2) \leq d_h(\boldsymbol{L}_0, \boldsymbol{L}_1) + d_h(\boldsymbol{L}_1, \boldsymbol{L}_2)$. In addition, we note that $d_h(c\boldsymbol{L}_0, c\boldsymbol{L}_1) = d_h(\boldsymbol{L}_0, \boldsymbol{L}_1) \; \forall c \in \mathbb{R}\backslash\{0\}$, emphasizing its interpretation as a fractional error.

This notion naturally extends to (positive semidefinite) graph Laplacians if one considers only vectors $\vec{x}$ that are orthogonal to their kernels (ie, require that $\vec{1}^{\intercal}\vec{x} = 0$ when taking the supremum in (17)). With this modification, the connection with the spectral graph sparsification can be stated as follows:

**Theorem 1.** *If $d_h\left(\boldsymbol{L}_G, \boldsymbol{L}_{\widetilde{G}}\right) \leq \ln(\sigma)$, then $\widetilde{G}$ is a $\sigma$-spectral approximation of $G$.*

Here, the notion of $\sigma$-spectral approximation is the same as in Spielman & Teng [6] (see Section 2), and thus is restricted to sparsification only. The proof is provided in Appendix Section D.

As $d_{\vec{x}}$ is analogous to the ratio of quadratic forms with $\vec{x}$, $d_h$ is likewise analogous to the notion of a $\sigma$-spectral approximation. Moreover, as $d_{\vec{x}}$ and $d_h$ also consider angular differences between $\boldsymbol{L}_G\vec{x}$ and $\boldsymbol{L}_{\widetilde{G},l}\vec{x}$, they serve as more sensitive measures of graph similarity.

In the following sections, we compare our algorithm with other graph reduction methods using $d_{\vec{x}}$, where we choose $\vec{x}$ to be eigenvectors of the original graph Laplacian. In Appendix Section H, we replicate our results using more standard measures (eg, quadratic forms and eigenvalues).

### 5.2 Comparison with spectral graph sparsification

Figure 2 compares our algorithm (prioritizing edge reduction, and excluding the possibility of contraction) with the standard spectral sparsification algorithm of Spielman & Srivastava [20] using three real-world datasets. We choose to compare with this particular sparsification method because it directly aims to optimally preserve the Laplacian. To the best of our knowledge, other sparsification methods either do not explicitly preserve properties associated with the Laplacian [46, 47], or share the same spirit as Spielman & Srivastava's algorithm [48] (often considering other settings, such as distributed [49] or streaming [50] computation). The results in Figure 2 show that our algorithm better preserves $\boldsymbol{L}^\dagger$ and preferentially preserves its action on eigenvectors associated with global structure.

### 5.3 Comparison with graph coarsening algorithms

Figure 3 compares our algorithm (prioritizing node reduction) with several existing coarsening algorithms using three more real-world datasets. In order to make a fair comparison with these existing methods, after contracting their prescribed groups of nodes, we appropriately lift the resulting reduced $\boldsymbol{L}_{\widetilde{G}}^\dagger$ (see Appendix Section C). We find that our algorithm more accurately preserves global structure.

## 6 Conclusion

In this work, we unify spectral graph sparsification and coarsening through the use of a single cost function that preserves the Laplacian pseudoinverse $\boldsymbol{L}^\dagger$. We describe a probabilistic algorithm for

graph reduction that employs edge deletion, contraction, and reweighting to keep $\mathbb{E}\big[\boldsymbol{L}_{\tilde{G}}^{\dagger}\big] = \boldsymbol{L}_G^{\dagger}$, and uses a new measure of edge importance ($\beta_\star$) to minimize its variance. Using synthetic and real-world datasets, we demonstrate that our algorithm more accurately preserves global structure compared to existing algorithms. We hope that our framework (or some perturbation of it) will serve as a useful tool for graph algorithms, numerical linear algebra, and machine learning.

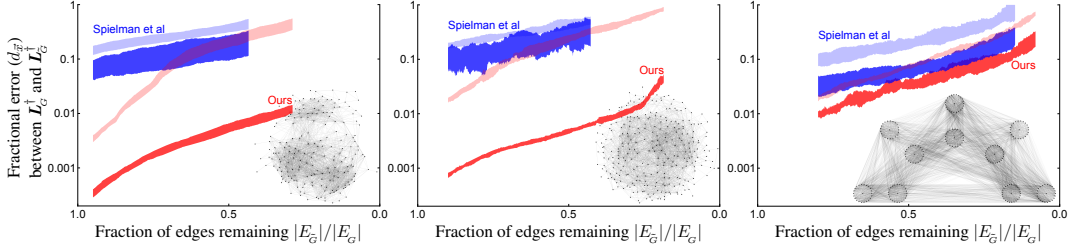

Figure 2: **Our sparsification algorithm preferentially preserves global structure**. We applied our algorithm without contraction (**Ours**) and compare with that of Spielman & Srivastava [20] (**Spielman et al**) using three datasets: *Left*: a collaboration network of Jazz musicians (198 nodes and 2742 edges) from [51]; *Middle*: the *C. elegans* posterior nervous system connectome (269 nodes and 2902 edges) from [52]; and *Right*: a weighted social network of face-to-face interactions between primary school students, with initial edge weights proportional to the number of interactions between pairs of students (236 nodes and 5899 edges) from [53]. For the two algorithms, we compute the hyperbolic distance $d_{\vec{x}}$ (fractional error) between $\boldsymbol{L}_G^{\dagger}\vec{x}$ and $\boldsymbol{L}_{\tilde{G}}^{\dagger}\vec{x}$ at different levels of sparsification for two choices of $\vec{x}$: the smallest non-trivial eigenvector of the original Laplacian (*dark shading*), which is associated with global structure; and the median eigenvector (*light shading*). Shading denotes one standard deviation about the mean for 16 runs of the algorithms. The curves end at the minimum edge density for which the sparsified graph is connected.

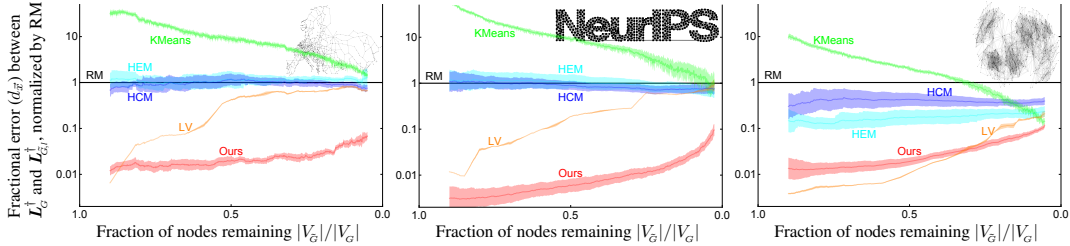

Figure 3: **Our algorithm preserves global structure more accurately than other coarsening algorithms**. We compare our algorithm (prioritizing node reduction) (**Ours**) to several existing coarsening algorithms: two classical methods for graph coarsening (heavy-edge matching (**HEM**) [54] and heavy-clique matching (**HCM**) [54]), and two recently proposed spectral coarsening algorithms (local variation by Loukas [55] (**LV**) and the $k$-means method by Jin & Jaja [21] (**KMeans**)). We ran the comparisons using three datasets: *Left*: a transportation network of European cities and roads between them (1039 nodes and 1305 edges) from [56]; *Middle*: a triangular mesh of the text "NeurIPS" (567 nodes and 1408 edges); and *Right*: a weighted social network of face-to-face interactions during an exhibition on infectious diseases, with initial edge weights proportional to the number of interactions between pairs of people (410 nodes and 2765 edges) from [57]. For all algorithms considered, we compute the hyperbolic distance $d_{\vec{x}}$ (fractional error) between $\boldsymbol{L}_G^{\dagger}\vec{x}$ and $\boldsymbol{L}_{\tilde{G},l}^{\dagger}\vec{x}$, where $\vec{x}$ is the smallest non-trivial eigenvector of the original Laplacian (associated with global structure). To provide a baseline, we plot their mean fractional error normalized by that obtained by random matching (**RM**) [54] for the same level of coarsening. Shading denotes one standard deviation about the mean for 16 runs of the algorithms.

**Acknowledgments**

We would like to thank Matthew de Courcy-Ireland for insightful discussions
and Ashlyn Maria Bravo Gundermsdorff for unique perspectives.

## Footnotes

[1]The authors agree with the sentiment of the footnote on page `xv` of [1], viz, omitting superfluous full stops to obtain a more efficient compression of, eg: *videlicet*, *exempli gratia*, etc.

[2]For animated examples using our graph reduction algorithm, see the following link:

[3] In the spirit of another common analogy (edge weights as conductances of a network of resistors), breaking a resistor is equivalent to deleting that edge, while contraction amounts to completely soldering over it.

[4]This expression is only officially applicable when the initial and final matrices are full-rank; additional care must be taken when they are not. However, for the case of changing the edge weights of a graph Laplacian, the original formula remains unchanged [37, 38] (so long as the graph remains connected), provided one uses the definitions in Section 3.5 (see also Appendix Sections C and F).

[5] $\boldsymbol{W}_n$ is often referred to as the "mass matrix" [40]. We note that the use of the random walk matrix $\boldsymbol{D}^{-1}\boldsymbol{L}$ can be seen as using the node degrees as a surrogate for the node weights.

[6]Specifically, randomly permute the nodes, and sequentially pair them with a random available neighbor (if there is one). The obtained set contains at least half as many edges as the maximum matching [45].

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
