[Supplementary Material]

# A Unifying Framework for Spectrum-Preserving Graph Sparsification and Coarsening: Appendix

## A Empirical validation of the approximation in equation (10)

In order to derive our graph reduction algorithm, we assume that the entries of the $M_e$ associated to different edges are approximately entrywise uncorrelated (Main Text Section 3.3). Similar to how the variance of the sum of independent random variables is the sum of their individual variances, this assumption allows us to approximate the expected squared Frobenius error of the final reduced graph $\mathbb{E}\big[\big\|L_{\widetilde{G}}^\dagger - L_G^\dagger\big\|_{\mathrm{F}}^2\big]$ as a sum over the sequence of probabilistic actions to individual edges:

$$\underbrace{\mathbb{E}\left[\left\|\sum \boldsymbol{\Delta L}^\dagger\right\|_{\mathrm{F}}^2\right]}_{\text{true error}} \approx \underbrace{\sum \mathbb{E}\left[\left\|\boldsymbol{\Delta L}^\dagger\right\|_{\mathrm{F}}^2\right]}_{\text{estimated error}}. \tag{18}$$

In Figure SI 1, we empirically validate this assumption for networks with a variety of structures. In fact, the true error is statistically equal to or less than the estimated error. Thus, the estimated error may be used by StopCriterion in Algorithm 1.

Figure SI 1: **The approximation of uncorrelated changes to $L^\dagger$ is nearly exact or a conservative estimate**. We test the validity of equation (18) using a variety of datasets: *Top left*: a triangular mesh of the text "NeurIPS" (567 nodes and 1408 edges); *Top middle*: an Erdős–Rényi model (256 nodes and $p = 1/16$); *Top right*: a weighted social network of face-to-face interactions between primary school students, with initial edge weights proportional to the number of interactions between pairs of students (236 nodes and 5899 edges) from [53]; *Bottom left*: a transportation network of European cities and roads between them (1039 nodes and 1305 edges) from [56]; *Bottom middle*: the *C. elegans* posterior nervous system connectome (269 nodes and 2902 edges) from [52]; and *Bottom right*: a collaboration network of Jazz musicians (198 nodes and 2742 edges) from [51]. We applied Algorithm 1, prioritizing edge reduction (allowing for deletion, contraction, and reweighting), and setting $q = 1/16$ and $d = 1/4$. We recorded the estimated error and the true error in $L^\dagger$ as a function of amount of reduction. Shading denotes one standard deviation about the mean for 32 runs of the algorithm. In general, the estimated error serves as an approximate upper bound of the true error in $L^\dagger$ (although it is nearly exact for graphs with a geometric quality). The validity of the approximation allows one to use a bound on the estimated error as a StopCriterion in Algorithm 1.

## B Derivation of the optimal probabilistic action to an edge

As discussed in Section 3.4, we seek to minimize:

$$\mathcal{C} = \mathbb{E}\left[\left\|\boldsymbol{\Delta L}^\dagger\right\|_{\mathrm{F}}^2\right] - \beta^2 \mathbb{E}[r], \tag{19}$$

subject to

$$\mathbb{E}\left[\boldsymbol{\Delta L}^{\dagger}\right] = \boldsymbol{0}. \tag{20}$$

When reducing multiple edges, $\mathbb{E}[r]$ is additive and $\mathbb{E}\left[\left\|\boldsymbol{\Delta L}^{\dagger}\right\|_{F}^{2}\right]$ is approximately additive (see Appendix Section A). Thus, we partition this minimization into a sequence of subproblems, treating each perturbation to an edge individually.

Recall that

$$\boldsymbol{\Delta L}^{\dagger} = \underbrace{f\left(\tfrac{\Delta w}{w_e}, w_e\Omega_e\right)}_{\text{nonlinear scalar}} \times \underbrace{\boldsymbol{M}_e,}_{\text{constant matrix}} \qquad \text{where} \qquad f = -\frac{\frac{\Delta w}{w_e}}{1 + \frac{\Delta w}{w_e}w_e\Omega_e}.$$

We now derive the optimal probability of deleting ($p_d$), contracting ($p_c$), or reweighting ($1 - p_d - p_c$) a given edge $e$, along with the change to its weight ($\Delta w$) in the case of the latter.

The constraint (20) requires that this reweight satisfies

$$\frac{p_d}{1 - w_e\Omega_e} - \frac{p_c}{w_e\Omega_e} + (1 - p_d - p_c)\,\mathbb{E}[f|\text{reweight}] = 0, \tag{21}$$

where we have used the following limits:

$$\begin{array}{llll} \text{deletion:} & \frac{\Delta w}{w_e} \to -1, & f \to (1 - w_e\Omega_e)^{-1} \\ \text{contraction:} & \frac{\Delta w}{w_e} \to +\infty, & f \to -(w_e\Omega_e)^{-1}. \end{array} \tag{22}$$

Likewise, the cost function (19) for acting on the edge $e$ becomes:

$$\mathcal{C} = \left(\frac{p_d}{(1 - w_e\Omega_e)^2} + \frac{p_c}{(w_e\Omega_e)^2} + (1 - p_d - p_c)\,\mathbb{E}\left[f^2|\text{reweight}\right]\right)m_e^2 - \beta^2\left(r_d p_d + r_c p_c\right), \tag{23}$$

where $r_d$ and $r_c$ are the number of prioritized items that would be removed by a deletion or contraction, respectively.

For a fixed $p_d$ and $p_c$, $\mathbb{E}[f|\text{reweight}]$ is fixed by equation (21). As $\frac{\partial^2 f}{\partial \Delta w^2} > 0$ everywhere, the inequality $\mathbb{E}\left[f^2|\text{reweight}\right] \geq \mathbb{E}[f|\text{reweight}]^2$ becomes an equality under minimization of (23).

Thus, if an edge is to be reweighted, it will be changed by the unique $\Delta w$ satisfying

$$\frac{p_d}{1 - w_e\Omega_e} - \frac{p_c}{w_e\Omega_e} - (1 - p_d - p_c)\frac{\frac{\Delta w}{w_e}}{1 + \frac{\Delta w}{w_e}w_e\Omega_e} = 0. \tag{24}$$

Clearly, the space of allowed solutions lies within the simplex $\mathcal{S}: 0 \leq p_d, 0 \leq p_c, p_d + p_c \leq 1$. The additional constraint $-1 \leq \frac{\Delta w}{w_e} \leq \infty$ further implies that $p_c \leq w_e\Omega_e$ and $p_d \leq 1 - w_e\Omega_e$. Hence, we substitute (24) into (23), and minimize it over this domain (given $m_e$, $w_e\Omega_e$, $\tau_e$, and $\beta$). After some careful elementary calculus, we obtain the solution provided in Figure 1 of the Main Text.

## C Lifting the matrices of a contracted graph

Here, we provide a detailed rationale for the definitions given in Section 3.5, namely, the choice of $\boldsymbol{L}_{\tilde{G}}$ and $\boldsymbol{L}_{\tilde{G}}^{\dagger}$, and how to "lift" these matrices to the original dimension $|V_G| \times |V_G|$ when edges have been contracted.

Recall the following definitions:

$$\boldsymbol{L}_{\tilde{G}} = \boldsymbol{W}_n^{-1}\boldsymbol{B}^{\top}\boldsymbol{W}_e\boldsymbol{B}, \tag{25}$$

$$\boldsymbol{L}_{\tilde{G}}^{\dagger} = \left(\boldsymbol{L}_{\tilde{G}} + \boldsymbol{J}\right)^{-1} - \boldsymbol{J}, \tag{26}$$

$$\boldsymbol{L}_{\tilde{G},l} = \boldsymbol{C}^{\top}\boldsymbol{L}_{\tilde{G}}\boldsymbol{W}_n^{-1}\boldsymbol{C}, \tag{27}$$

$$\boldsymbol{L}_{\tilde{G},l}^{\dagger} = \boldsymbol{C}^{\top}\boldsymbol{L}_{\tilde{G}}^{\dagger}\boldsymbol{W}_n^{-1}\boldsymbol{C}, \tag{28}$$

where

$$\boldsymbol{J} = \frac{1}{\vec{1}^{\top}\vec{w}_n}\vec{1}\vec{w}_n^{\top}, \tag{29}$$

$$\boldsymbol{C} = \{c_{ij}\} = \begin{cases} 1 & \text{node } j \text{ in supernode } i \\ 0 & \text{otherwise.} \end{cases} \tag{30}$$

The above definitions ensure that the lifted $\boldsymbol{L}_{\tilde{G},l}^{\dagger}$ of the contracted graph is identical to the $w_e \to \infty$ limit of the original $\boldsymbol{L}_G^{\dagger}$.

Figure SI 2: **Contracting the center edge of a line graph**. *Left:* Original graph $G$ with large weight $w_e$ on the center edge. *Right:* Reduced graph $\widetilde{G}$ obtained by contracting this edge ($w_e \to \infty$). Note that the weight of the contracted nodes sum to give the weight of the resulting supernode in the reduced graph.

To illustrate the consistency of these definitions, we consider a concrete example: the line graph with 3 edges, where the center edge is to be contracted (Figure SI 2). Let the center edge have weight $w_e \gg 1$, while the other two have a fixed weight of $1$.

For the original graph $G$, the Laplacian and its pseudoinverse are

$$
L_G = \begin{pmatrix} 1 & -1 & 0 & 0 \\ -1 & 1+w_e & -w_e & 0 \\ 0 & -w_e & 1+w_e & -1 \\ -0 & 0 & -1 & 1 \end{pmatrix}, \quad L_G^\dagger = \frac{1}{8} \begin{pmatrix} 5+\frac{2}{w_e} & -1+\frac{2}{w_e} & -1-\frac{2}{w_e} & -3-\frac{2}{w_e} \\ -1+\frac{2}{w_e} & 1+\frac{2}{w_e} & 1-\frac{2}{w_e} & -1-\frac{2}{w_e} \\ -1-\frac{2}{w_e} & 1-\frac{2}{w_e} & 1+\frac{2}{w_e} & -1+\frac{2}{w_e} \\ -3-\frac{2}{w_e} & -1-\frac{2}{w_e} & -1+\frac{2}{w_e} & 5+\frac{2}{w_e} \end{pmatrix}.
$$

For the contracted graph $\widetilde{G}$, we have

$$
W_n = \begin{pmatrix} 1 & 0 & 0 \\ 0 & 2 & 0 \\ 0 & 0 & 1 \end{pmatrix}, \quad J = \begin{pmatrix} \frac{1}{4} & \frac{1}{2} & \frac{1}{4} \\ \frac{1}{4} & \frac{1}{2} & \frac{1}{4} \\ \frac{1}{4} & \frac{1}{2} & \frac{1}{4} \end{pmatrix}, \quad C = \begin{pmatrix} 1 & 0 & 0 & 0 \\ 0 & 1 & 1 & 0 \\ 0 & 0 & 0 & 1 \end{pmatrix}.
$$

Thus, the reduced Laplacian and its pseudoinverse are

$$
L_{\widetilde{G}} = \begin{pmatrix} 1 & -1 & 0 \\ -\frac{1}{2} & 1 & -\frac{1}{2} \\ 0 & -1 & 1 \end{pmatrix}, \quad L_{\widetilde{G}}^\dagger = \frac{1}{8} \begin{pmatrix} 5 & -2 & -3 \\ -1 & 2 & -1 \\ -3 & -2 & 5 \end{pmatrix}.
$$

When lifted to the original dimensions $|V_G| \times |V_G|$, these become

$$
L_{\widetilde{G},l} = \begin{pmatrix} 1 & -\frac{1}{2} & -\frac{1}{2} & 0 \\ -\frac{1}{2} & \frac{1}{2} & \frac{1}{2} & -\frac{1}{2} \\ -\frac{1}{2} & \frac{1}{2} & \frac{1}{2} & -\frac{1}{2} \\ 0 & -\frac{1}{2} & -\frac{1}{2} & 1 \end{pmatrix}, \quad L_{\widetilde{G},l}^\dagger = \frac{1}{8} \begin{pmatrix} 5 & -1 & -1 & -3 \\ -1 & 1 & 1 & -1 \\ -1 & 1 & 1 & -1 \\ -3 & -1 & -1 & 5 \end{pmatrix}.
$$

Note that the lifted $L_{\widetilde{G},l}^\dagger$ is equal to the $w_e \to \infty$ limit of the original $L_G^\dagger$, as desired. In contrast, the original $L_G$ diverges, while the lifted $L_{\widetilde{G},l}$ averages the rows and columns of the merged nodes. Moreover, regardless of whether node weights are included in the definitions, using the standard Moore–Penrose pseudoinverse of the reduced Laplacian will yield a lifted pseudoinverse that is not equivalent to the original in the $w_e \to \infty$ limit.

Additionally, we remark that, while contraction always requires the summing of node weights, it can also lead to the summing of edge weights (when the contracted edge participates in any triangle in the original graph, see Figure SI 3).

# D Proof of the relationship between the hyperbolic distance and $\sigma$-spectral approximation

In this section, we prove Theorem 1 from Section 5.1.

Figure SI 3: **Contracting an edge that participates in triangles**. *Left:* Original graph $G$ containing an edge with large weight $w_e$ that participates in two triangles. *Right:* Reduced graph $\widetilde{G}$ obtained by contracting this edge ($w_e \to \infty$). Note that the two non-contracted edges in each triangle form a single edge in the reduced graph with weight equal to their sum.

**Theorem 1.** *If $d_h\left(\boldsymbol{L}_G, \boldsymbol{L}_{\widetilde{G}}\right) \le \ln(\sigma)$, then $\widetilde{G}$ is a $\sigma$-spectral approximation of $G$.*

*Proof.* Let $G$ be the original graph and $\widetilde{G}$ its sparse approximation (no contraction/removing of nodes). Recall the relevant definitions:

$\widetilde{G}$ is a $\sigma$-spectral approximation of $G$ [6] if

$$\frac{1}{\sigma}\vec{x}^\top \boldsymbol{L}_G \vec{x} \le \vec{x}^\top \boldsymbol{L}_{\widetilde{G}} \vec{x} \le \sigma \vec{x}^\top \boldsymbol{L}_G \vec{x}, \quad \forall \vec{x} \in \mathbb{R}^{|V_G|}. \tag{31}$$

We propose to instead measure the hyperbolic distance between the resulting $\boldsymbol{L}_G \vec{x}$ and $\boldsymbol{L}_{\widetilde{G}} \vec{x}$, namely

$$d_h\left(\boldsymbol{L}_G, \boldsymbol{L}_{\widetilde{G}}\right) \overset{\text{def}}{=} \sup_{\vec{x} \perp \vec{1}} \left\{ \operatorname{arccosh}\left(1 + \frac{\left\|(\boldsymbol{L}_G - \boldsymbol{L}_{\widetilde{G}})\vec{x}\right\|_2^2 \|\vec{x}\|_2^2}{2\left(\vec{x}^\top \boldsymbol{L}_G \vec{x}\right)\left(\vec{x}^\top \boldsymbol{L}_{\widetilde{G}} \vec{x}\right)}\right) \right\}, \tag{32}$$

where $\boldsymbol{L}_G$ and $\boldsymbol{L}_{\widetilde{G}}$ are the Laplacians of $G$ and $\widetilde{G}$, respectively, and $\vec{x}$ is perpendicular to their kernels.

Consider the result of a Laplacian acting on such a vector $\vec{x}$, and decompose the output as a component parallel to $\vec{x}$ with magnitude $\ell_\parallel$ and a component $\vec{\ell}_\perp$ perpendicular to $\vec{x}$:

$$\boldsymbol{L}_G \vec{x} = \widetilde{\ell}_\parallel \frac{\vec{x}}{\|\vec{x}\|_2} + \vec{\ell}_\perp, \qquad \boldsymbol{L}_{\widetilde{G}} \vec{x} = \widetilde{\ell}_\parallel \frac{\vec{x}}{\|\vec{x}\|_2} + \vec{\widetilde{\ell}}_\perp. \tag{33}$$

Hence,

$$\left\|(\boldsymbol{L}_G - \boldsymbol{L}_{\widetilde{G}})\vec{x}\right\|_2^2 = (\ell_\parallel - \widetilde{\ell}_\parallel)^2 + \left\|\vec{\ell}_\perp - \vec{\widetilde{\ell}}_\perp\right\|_2^2, \tag{34}$$

$$\vec{x}^\top \boldsymbol{L}_G \vec{x} = \ell_\parallel \|\vec{x}\|_2, \qquad \vec{x}^\top \boldsymbol{L}_{\widetilde{G}} \vec{x} = \widetilde{\ell}_\parallel \|\vec{x}\|_2. \tag{35}$$

Let $z = \widetilde{\ell}_\parallel / \ell_\parallel$. Substituting (35) into (31), we see that $\widetilde{G}$ is a $\sigma$-spectral approximation of $G$ if

$$\max\left\{\frac{\widetilde{\ell}_\parallel}{\ell_\parallel}, \frac{\ell_\parallel}{\widetilde{\ell}_\parallel}\right\} = \max\left\{z, \frac{1}{z}\right\} \le \sigma. \tag{36}$$

Now, substituting (34) into (16), we obtain:

$$d_{\vec{x}}\left(\boldsymbol{L}_G, \boldsymbol{L}_{\widetilde{G}}\right) = \operatorname{arccosh}\left(1 + \frac{\left((\ell_\parallel - \widetilde{\ell}_\parallel)^2 + \left\|\vec{\ell}_\perp - \vec{\widetilde{\ell}}_\perp\right\|_2^2\right)\|\vec{x}\|_2^2}{2\ell_\parallel \|\vec{x}\|_2 \widetilde{\ell}_\parallel \|\vec{x}\|_2}\right)$$

$$\ge \operatorname{arccosh}\left(1 + \frac{\widetilde{\ell}_\parallel^2 - 2\widetilde{\ell}_\parallel \ell_\parallel + \ell_\parallel^2}{2\ell_\parallel \widetilde{\ell}_\parallel}\right)$$

$$\ge \operatorname{arccosh}\left(\frac{1}{2}\left(z + \frac{1}{z}\right)\right).$$

Using the identity $\operatorname{arccosh}(x) = \ln\left(x + \sqrt{x^2 - 1}\right)$,

$$d_{\vec{x}}\left(\boldsymbol{L}_G, \boldsymbol{L}_{\widetilde{G}}\right) \ge \ln\left(\frac{1}{2}\left(z + \frac{1}{z}\right) + \sqrt{\frac{1}{4}\left(z + \frac{1}{z}\right)^2 - 1}\right)$$

$$\ge \ln\left(\frac{1}{2}\left(z + \frac{1}{z}\right) + \frac{1}{2}\left|z - \frac{1}{z}\right|\right)$$

$$\ge \left|\ln(z)\right|.$$

Thus, if $d_{\vec{x}}\big(\boldsymbol{L}_G, \boldsymbol{L}_{\widetilde{G}}\big) \leq \ln(\sigma)\ \forall \vec{x} \perp \vec{1}$, then $\widetilde{G}$ is a $\sigma$-spectral approximation of $G$, as desired. $\qquad\square$

# E  Number of edges acted upon per iteration can be $\mathcal{O}(|V|)$

In this section, we study the effect of varying the parameter $q$, the fraction of sampled edges acted upon, using real-world datasets from different domains (Figure SI 4).

For each iteration of our algorithm, we sample a random independent edge set and act on the fraction $q$ with the lowest $\beta_{\star e}$ (see Main Text Section 4). We find that the resulting error asymptotes around $q \sim 1/16$. We expect that by combining this sampling method with existing algorithmic primitives (eg, [20], see Appendix Section F), our algorithm could achieve a running time of $\widetilde{\mathcal{O}}(\langle k \rangle |E|)$, where $\langle k \rangle$ is the average degree (see Main Text Section 4). This would allow it to be used in large-scale applications of graph reduction.

Figure SI 4: **Number of sampled edges acted upon per iteration can be $\mathcal{O}(|V|)$.** We study the effect of varying $q$, the fraction of the sampled edges that are acted upon per iteration, using three datasets: *Left*: a transportation network of European cities and roads between them (1039 nodes and 1305 edges) from [55]; *Middle*: the *C. elegans* posterior nervous system connectome (269 nodes and 2902 edges) from [52]; and *Right*: a weighted social network of face-to-face interactions during an exhibition on infectious diseases, with initial edge weights proportional to the number of interactions between pairs of people (410 nodes and 2765 edges) from [57]. We prioritize edge reduction (allowing for deletion, contraction, and reweighting). At each iteration, the algorithm randomly samples a maximal independent edge set, and chooses $\beta$ such that a fraction $q$ of these edges (with the lowest $\beta_{\star e}$) are acted upon. For each run, we compute the hyperbolic distance $d_{\vec{x}}$ (fractional error) between $\boldsymbol{L}_G^{\dagger}\vec{x}$ and $\boldsymbol{L}_{\widetilde{G},l}^{\dagger}\vec{x}$, where $\vec{x}$ is one of three eigenvectors of the original Laplacian. *Top* plots display the results when the graph has $1/2$ of its original number of edges, and *bottom* plots when it has $1/12$. Shading denotes one standard deviation about the mean for 8 runs of the algorithm for a given value of $q$. Note that a significant fraction ($q \sim 1/16$) of the sampled edges can be reduced each iteration without sacrificing much in terms of accuracy. As, empirically, the size of the independent edge sets are typically $\mathcal{O}(|V|)$, the number of edges acted upon per iteration can likewise be $\mathcal{O}(|V|)$.

# F  Efficiently computing $m_e$

As discussed in Main Text Section 4, the main computational bottleneck of our algorithm is computing $\Omega_e$ and $m_e$. For $\Omega_e$, we can draw on the work of [20], which describes a method for efficiently computing $\varepsilon$-approximate values of $\Omega_e$ for all edges, requiring $\widetilde{\mathcal{O}}(|E| \log |V|/\epsilon^2)$ time. In this section, we describe an analogous procedure to efficiently compute the $m_e$.

Recall that the reduced Laplacian is:

$$L_{\widetilde{G}} = W_n^{-1} B^\top W_e B,$$

hence, the quantity $\widehat{L_{\widetilde{G}}} \overset{\text{def}}{=} W_n^{1/2} L_{\widetilde{G}} W_n^{-1/2}$ is clearly symmetric.

Less obvious is the fact that $\widehat{L_{\widetilde{G}}^\dagger} \overset{\text{def}}{=} W_n^{1/2} L_{\widetilde{G}}^\dagger W_n^{-1/2}$ is also symmetric. This can be seen by noting that $W_n^{1/2} J W_n^{-1/2}$ is symmetric, and using the definition of the inverse (equation (26)):

$$
\begin{aligned}
\widehat{L_{\widetilde{G}}^\dagger} &= W_n^{1/2} L_{\widetilde{G}}^\dagger W_n^{-1/2} \\
&= W_n^{1/2} \left( \left( L_{\widetilde{G}} + J \right)^{-1} - J \right) W_n^{-1/2} \\
&= W_n^{1/2} \left( L_{\widetilde{G}} + J \right)^{-1} W_n^{-1/2} - W_n^{1/2} J W_n^{-1/2} \\
&= \left( W_n^{1/2} L_{\widetilde{G}} W_n^{-1/2} + W_n^{1/2} J W_n^{-1/2} \right)^{-1} - W_n^{1/2} J W_n^{-1/2}.
\end{aligned}
$$

We also remark that $\widehat{L_{\widetilde{G}}^\dagger}$ is indeed the pseudoinverse of $\widehat{L_{\widetilde{G}}}$:

$$\widehat{L_{\widetilde{G}}^\dagger} \widehat{L_{\widetilde{G}}} = \widehat{L_{\widetilde{G}}} \widehat{L_{\widetilde{G}}^\dagger} = I - W_n^{1/2} J W_n^{-1/2}$$

The change to the reduced Laplacian $L_{\widetilde{G}}$ is given by

$$\Delta L_{\widetilde{G}} = W_n^{-1} \vec{b}_e \Delta w_e \vec{b}_e^\top$$

Thus, by the Woodbury matrix identity, the change to its inverse is

$$\Delta L_{\widetilde{G}}^\dagger = f w_e L_{\widetilde{G}}^\dagger W_n^{-1} \vec{b}_e \vec{b}_e^\top L_{\widetilde{G}}^\dagger$$

where $f$ is given by equation (5).

Lifting this change back to the original dimension via equation (28) gives

$$\Delta L_{\widetilde{G},l}^\dagger = f w_e C^\top L_{\widetilde{G}}^\dagger W_n^{-1} \vec{b}_e \vec{b}_e^\top L_{\widetilde{G}}^\dagger W_n^{-1} C$$

In particular, as $L_{\widetilde{G}}^\dagger W_n^{-1}$ is symmetric, $\Delta L_{\widetilde{G},l}^\dagger$ is also symmetric, thus we can write the Frobenius norm as

$$\left\| \Delta L_{\widetilde{G},l}^\dagger \right\|_F = f w_e \vec{b}_e^\top L_{\widetilde{G}}^\dagger W_n^{-1} C C^\top L_{\widetilde{G}}^\dagger W_n^{-1} \vec{b}_e \tag{37}$$

$$= f m_e \tag{38}$$

Note that the definition of $m_e$ provided in Section 3.3 of the Main Text (equation (9)) applies to the case of unit node weights, and the general expression is given by

$$m_e = w_e \vec{b}_e^\top L_{\widetilde{G}}^\dagger L_{\widetilde{G}}^\dagger W_n^{-1} \vec{b}_e, \tag{39}$$

where we have used $C C^\top = W_n$.

Thus, we can express $m_e$ in terms of $\widehat{L_{\widetilde{G}}^\dagger}$:

$$
\begin{aligned}
m_e &= w_e \vec{b}_e^\top W_n^{-1/2} \widehat{L_{\widetilde{G}}^\dagger} \widehat{L_{\widetilde{G}}^\dagger} W_n^{-1/2} \vec{b}_e \\
&= w_e \left\| \widehat{L_{\widetilde{G}}^\dagger} W_n^{-1/2} \vec{b}_e \right\|_2^2.
\end{aligned}
$$

We can now use the Johnson–Lindenstrauss lemma to build a structure from which one can efficiently compute approximations of $m_e$. Let $Q$ be a random projection matrix of size $k \times n$, where $k = \mathcal{O}(\log n / \varepsilon^2)$, then one can compute $\varepsilon$-approximations of $m_e$ as follows:

$$m_e \approx w_e \left\| Q \widehat{L_{\widetilde{G}}^\dagger} W_n^{-1/2} \vec{b}_e \right\|_2^2.$$

Let $Z = Q \widehat{L_{\widetilde{G}}^\dagger}$, and denote the $i^{\text{th}}$ rows of $Q$ and $Z$ by $\vec{q}_i$ and $\vec{z}_i$, respectively. Then, one can make $k$ calls to an efficient algebraic multigrid solver (we used the *pyamg* package [58]) to obtain approximate solutions to $\widehat{L_{\widetilde{G}}} \vec{z}_i = \vec{q}_i$ for the $k$ rows of $Z$. An approximation to the $m_e$ of any edge can now be computed by taking the difference between the columns of $Z W_n^{-1/2}$ corresponding to the two nodes jointed by this edge, and taking the squared 2-norm of the result.

## F.1 Constructing the projection matrix

Care must be taken in constructing the projection matrix $\boldsymbol{Q}$. In particular, its rows must be orthogonal to the null space of $\widehat{\boldsymbol{L}_{\widetilde{G}}}$, namely $\vec{w}_n^{1/2}$. In addition, the columns must be nearly unit length. To this end, we initialize $\boldsymbol{Q}$ as a random matrix with entries $\{1/\sqrt{k}, -1/\sqrt{k}\}$ with equal probability and iterate the following steps:

1. For each column, scale its values such that it has unit length
2. For each row, subtract its weighted mean $\vec{q}_i^\top \vec{w}_n^{1/2} / \vec{1}^\top \vec{w}_n^{1/2}$

We iterate this procedure until the columns have nearly unit lengths, to within a factor sufficiently smaller than $\varepsilon$.

As a proof of concept, in Figure SI 5, we show the approximate $m_e$ as a function of their exact values.

Figure SI 5: **Efficient approximation of** $m_e$. As a proof of concept, we compare the approximation of $m_e$ (computed using the procedure described in this section) with their exact values. Here, we consider a $64 \times 64$ torus graph (4096 nodes and 8192 edges), where the edge weights are randomly distributed as $\exp(U(-2,2))$, where $U(a,b)$ is the uniform distribution. To calculate the approximate $m_e$, we project from 4096 to 33 dimensions, resulting in approximations that are typically within a factor of 1.27 of the exact value.

## G   Perturbations to eigenvalues of the Laplacian pseudoinverse

We first provide the lowest order change in the eigenvalues of $\boldsymbol{L}_G^\dagger$. Then, we show how it relates to the Frobenius norm of the perturbation, explicitly relating it to our graph reduction algorithm.

Consider an inverse Laplacian $\boldsymbol{L}^\dagger$, which has an eigenvector $\vec{x}$ (without loss of generality, assume $\|\vec{x}\|_2 = 1$) with associated eigenvalue $\lambda$. If we perturb $\boldsymbol{L}^\dagger$ by $\varepsilon \boldsymbol{\Delta L}^\dagger$, we can solve for the first-order corrections to this "eigenpair" as follows:

$$(\boldsymbol{L}^\dagger + \varepsilon \boldsymbol{\Delta L}^\dagger)(\vec{x} + \varepsilon \Delta \vec{x}) = (\lambda + \varepsilon \Delta \lambda)(\vec{x} + \varepsilon \Delta \vec{x})$$
$$(\boldsymbol{L}^\dagger - \lambda)\Delta \vec{x} = (\Delta \lambda - \boldsymbol{\Delta L}^\dagger)\vec{x} + \mathcal{O}(\varepsilon),$$

where we have used $\boldsymbol{L}^\dagger \vec{x} = \lambda \vec{x}$.

Taking the inner product with $\vec{x}$ gives

$$\vec{x}^\top(\boldsymbol{L}^\dagger - \lambda)\Delta \vec{x} = \vec{x}^\top(\Delta \lambda - \boldsymbol{\Delta L}^\dagger)\vec{x}$$
$$\Delta \vec{x}^\top(\boldsymbol{L}^\dagger - \lambda)\vec{x} = \Delta \lambda \vec{x}^\top \vec{x} - \vec{x}^\top \boldsymbol{L}^\dagger \vec{x}$$
$$0 = \Delta \lambda - \vec{x}^\top \boldsymbol{L}^\dagger \vec{x},$$

where we have used the symmetry of $\boldsymbol{L}^\dagger$.

This provides the first-order correction to the eigenvalues of $\boldsymbol{L}^\dagger + \varepsilon \boldsymbol{\Delta L}^\dagger$:

$$\Delta \lambda = x^\top \boldsymbol{\Delta L}^\dagger \vec{x}. \tag{40}$$

The correction in (40) is controlled by the operator norm of $\boldsymbol{\Delta L}^\dagger$,

$$\Delta \lambda = \vec{x}^\top \boldsymbol{\Delta L}^\dagger \vec{x} \le \sup_{\|\vec{x}\|_2 = 1} \left\| \boldsymbol{\Delta L}^\dagger \vec{x} \right\|_2 = \left\| \boldsymbol{\Delta L}^\dagger \right\|_{\text{op}}.$$

Thus, bounding the first-order correction to the eigenvalues,

$$|\Delta\lambda| \le \left\| \boldsymbol{\Delta L}^\dagger \right\|_{\text{op}}. \tag{41}$$

As the operator norm is bounded by the Frobenius norm (by the Cauchy–Schwarz inequality), the estimated error (ie, $\sum \mathbb{E}\left[\left\| \boldsymbol{\Delta L}^\dagger \right\|_{\text{F}}^2\right]$, equation (18)) provides a conservative bound for the change in the eigenvalues of the resulting reduced graph.

Moreover, as the bound is the same for all eigenvalues of the perturbed $\boldsymbol{L}^\dagger$, the *relative* error is more tightly bounded for its largest eigenvalues (those associated with large-scale structure).

# H Comparison of graph reduction methods using typical similarity measures

Our proposed hyperbolic distance is not usually used as a measure of similarity. Hence, in this section, we show that other more commonly used measures yield similar results when comparing graph reduction algorithms.

## H.1 Sparsification

Figure SI 6 compares our algorithm (prioritizing edge reduction, and excluding the possibility of contraction) with the spectral sparsification algorithm of [20] using a stochastic block model (SBM) with four distinct communities. We choose a highly associative SBM due to the clear separation between the eigenvectors associated with global structure (ie, the communities) and the bulk of the spectrum. Note that these algorithms have different objectives (preserving $\boldsymbol{L}^\dagger$ and $\boldsymbol{L}$, respectively), and both accomplish their desired goal.

Figure SI 6: **Our sparsification algorithm preferentially preserves global structure**. We compare our algorithm without contraction (in **red**) with that of Spielman & Srivastava [20] (in **blue**) using a symmetric stochastic block model (256 nodes, 4 communities, and intra- and inter-community connection probabilities of $2^{-2}$ and $2^{-6}$, respectively). We ran both algorithms 16 times on the same initial graph. For each eigenvector of the original Laplacian, we compute the mean and standard deviation of its quadratic forms (with $\boldsymbol{L}_{\tilde{G}}$ and with $\boldsymbol{L}_{\tilde{G}}^\dagger$) as a function of edges remaining. We divide the eigenvectors into two groups: the 3 nontrivial eigenvectors ("global structure") and the remaining eigenvectors ("local details"), and compute the average mean and average standard deviation for each group. Shading denotes one (average) standard deviation about the (average) mean. *Left*: Laplacian pseudoinverse quadratic form. *Right*: Standard Laplacian quadratic form. Note that the upward bias of the "reciprocal" quadratic form is expected for both algorithms (as $\mathbb{E}[X] \le 1/\mathbb{E}[1/X]$ for any random variable $X > 0$).

## H.2 Coarsening

Figure SI 7 replicates the results of Figure 3, but uses the Laplacian pseudoinverse quadratic form to measure the reduction quality instead of our proposed hyperbolic distance.

Figure SI 8 compares our method with that of Loukas [55], using the average relative error of the $k$ lowest non-trivial eigenvalues of the Laplacian (ie, $\frac{1}{k}\sum_{i=2}^{k+1} \left|\widetilde{\lambda}_i - \lambda_i\right|/\lambda_i$) to measure the reduction quality.

Figure SI 7: **Our coarsening algorithm performs even better when using the quadratic form with $\boldsymbol{L}^{\dagger}$**. Here we replicate the experiments in Figure 3. However, instead of using our proposed hyperbolic distance, we consider the logarithm of the fractional change in the Laplacian pseudoinverse quadratic form for $\vec{x}$ the lowest non-trivial eigenvector of the original Laplacian: $\left|\log\left(\vec{x}^{\top}\boldsymbol{L}_{\tilde{G}}^{\dagger}\vec{x}/\vec{x}^{\top}\boldsymbol{L}_{G}^{\dagger}\vec{x}\right)\right|$. As before, for each algorithm, we plot the mean of this quantity normalized by that obtained by random matching (**RM**). Shading denotes one standard deviation about the mean for 16 runs of the algorithms. The results are remarkably similar to those obtained using our proposed hyperbolic distance (Figure 3). The most notable deviation is that our algorithm appears to perform *better* when compared using this quadratic form.

Figure SI 8: **Our algorithm preferentially preserves the lower portion of the Laplacian spectrum**. We compare our coarsening algorithm (**Ours**) with that of Loukas [55] (**LV**) using the same three datasets as in Figure 3. We use the relative error in the $k$ lowest non-trivial eigenvalues of the Laplacian: $\frac{1}{k}\sum_{i=2}^{k+1}\left|\widetilde{\lambda}_i - \lambda_i\right|/\lambda_i$, a measure of spectral similarity considered in [55]. Shading denotes one standard deviation about the mean for 8 runs of the algorithms. Note that our algorithm performs considerably better when applied to graphs with a geometric quality.

# I   Applications to graph visualization

Data visualization is an important (and aesthetically pleasing) application of graph reduction. As such, we generated videos of our algorithm reducing several real-world datasets. Figure SI 9 displays several stages of our algorithm applied to a temporal social network. A video of this reduction can be found here; an application to an airport network (a case with both geometric and scale-free aspects) can be found here; an application to the European road network can be found here, and a reduction of a "hierarchical meta-graph" can be found here.[1]

youtube.com/watch?v=qqLJclVUML8; youtube.com/watch?v=tXUr6RBRaEI;
youtube.com/watch?v=UVhT0y4Uae0; and youtube.com/watch?v=i3u4kkxMK40.

Figure SI 9: **Visualization of our graph reduction algorithm preserving global structure**. We applied our algorithm (prioritizing edge reduction, and allowing for deletion, contraction, and reweighting) to a weighted social network of face-to-face interactions during an exhibition on infectious diseases, with initial edge weights proportional to the number of interactions between pairs of people (410 nodes and 2765 edges) from [57]. Node color indicates the lowest nontrivial eigenvector of the reduced Laplacian, which in this case is aligned with the temporal direction. This graph displays a notable amount of hierarchical clustering (owing to its social nature), which is reflected in the reduced graphs. Eg, our algorithm begins by collapsing small, tightly-knit clusters of several people into one "supernode", corresponding to groups of people who visited the exhibition together. A video of this reduction can be found here.

$|V| = 410$

$|E| = 2765$

$|V| = 17$

$|E| = 25$

Figure 3: Fractional error in the inverse Laplacian, $\|L_{\widetilde{G}}^{\dagger}x - L_G^{\dagger}x\|_2 / \|L_G^{\dagger}x\|_2$. **Left:** Spielman and collaborators' sparsification method. **Right:** Our method, using both deletion and contraction.

Let $B$ be the $|E| \times |V|$ signed incidence matrix, and $W_e$ ($W_n$) be the diagonal matrix of edge (node) weights. The familiar graph Laplacian is given by $L = B^\top W_e B$, but this tacitly assumes that the nodes are identically important. Differential geometry offers a prescription for how to incorporate node weights into the Laplacian; treating a graph as a simplicial complex, the Hodge Laplacian for 0-forms (functions on vertices) is given by $\delta d$, where the differential $d = B$, and the codifferential $\delta = W_n^{-1}B^\top W_e$. Thus, in cases where the nodes have an additive measure of importance, it is appropriate to use $L = W_n^{-1}B^\top W_e B$. To obtain the effective Laplacian in the original node basis, use the projection matrices: $C^\top L W_n^{-1} C$.

## Footnotes

[1]Explicit urls for the non-hyperlinked:

### References

[1] I. Lorenzo, et al. What's in a Crowd? Analysis of Face-to-Face Behavioral Networks. *J. of Theoretical Biology*, 271(1):166–180, 2011