[Reviews · NeurIPS 2019]

Reviewer 1



Originality The paper seems quite original, with new algorithms based on a novel principle of similarity between graphs. Quality In general, the developed framework is not sufficiently analyzed. This paper contains no error bounds on the similarity between the reduced and original pseudoinverse of Laplacian, nor any results in this direction. There is also no formal reasoning as to why minimizing ||\Delta L^{-1}||_F should particularly result in preservation of the top eigenvectors. There is no sense given of how sparse the graph can be made while keeping ||\Delta L^{-1}||_F small. There are many hyperparameters to Algorithms 1 and 2, and their choice is only loosely discussed as a purely empirical matter. The experiments are well flushed out and strong. However, there is no empirical evaluation of simultaneous reduction of nodes and edges, a main advertisement of the paper. Clarity Although the writing flowed well, I found this paper relatively confusing. It is not made sufficiently clear / precise why preservation (in Frobenius norm) of the psuedoinverse of the Laplacian is desired. For instance: [75 - 76] Spectral sparsification preserves the solution x* of Lx = b in a certain sense (see e.g. the monograph “Lx = b”). Does preservation of L^{-1} in the Frobenius norm imply this same property? Does it imply preservation in a different sense? [81 - 82] Can the statement that the 2nd eigenvector will be “preferentially preserved” be elaborated on? This feels to me like an interesting statement, and it comes up again in the experiments, but is discussed / justified in broad terms only. The motivation for the paper is also somewhat unclear. It is not clear whether preservation of L^{-1} is a natural framework in which to consider graph coarsening/sparsification, or whether it is merely a convenient framework. E.g. why is this considered as opposed to the “ ‘restricted’ Laplacian quadratic form” mentioned in [55-57]? I am also confused about the desire for unbiasedness in section 3, i.e. the requirement that E(L_{tilde{G}})^{-1}) = E(L_G^{-1}). This seems to be a somewhat arbitrary desire, as opposed to data reduction and preservation of ||\Delta L^{-1}||_F. Is it made purely to simplify the derivations (9) - (11), or is there a claim that it is truly a desirable property? Additionally, I think 3.4 could be reworked. It is not clear that there is a strong takeaway, and in the absence of this, this section contains a lot of implicit mathematical manipulations without much message. It is also confusing, for instance what is B_{1d} (or B_{1c}, or B_2) when one is interested in reducing both the number of nodes and edges? Similarly, 3.5 seems unnecessary. It is also not stated explicitly how the multi-step scheme eliminates the ‘undesirable behavior’ of the single-step algorithm . Significance The paper seems to tackle a relevant issue (graph compression) in a novel way, with strong empirical results. However, without more of a theoretical grounding, and more details on how to implement the given algorithm (for instance, how to choose a stop criterion in Algorithm 2), it is difficult to see it being of high practical significance.

Reviewer 2



The paper considers a framework to coarsen (and/or sparsify) a graph via a probabilistic scheme that tries to preserve the spectral properties of the Laplacian pseudoinverse. The authors demonstrate how this viewpoint can provide a 'unifying' framework from which the contraction of nodes (coarsening) as well as the removal of edges can be considered. I think this is a very interesting paper, even though I have a number of concerns as listed below. Quality ======= Overall the claims the authors make are well supported. However, ocassionally their arguments feel a bit imprecise. In section 3.2, for eq 5-6: the authors argue that edges are uncorrelated is a 'reasonable assumption, when one is considering a 'small fraction of edges'. There are several issues here: first, I feel their should be more explanation than a half sentence in parenthesis of why this should hold in such a limit. Second, they consider quite large fractions of edges afterwards (see e.g. Figure 2). Third it will certainly also depend on the position of the edges whether they are correlated or not. Some of these effects the authors account for themselves in their multi-step scheme when they discuss the merging of two important nodes and one unimportant node. The algorithm the authors propose starts from the full graph and then eliminates edges. Now, if the argument is that the graph is too large to be kept in memory to start with, seems to be not a very good option, and at least slow down the first few iterations -- is there any way to adjust their method to act more local or start from an empty graph? This would merit more discussion here. What is the computational complexity of the algorithm? Can the authors say something about the numerical stability of the Sherman-Woodbury-Morrison updates? In general I think these updates do not have to be numericall stable. Sampling edges uniformly at random -- is this done for convenience or is there some kind of optimality associated with this sampling scheme? (e.g. in contrast in sparsification edges are also sampled according to the ratio of effective resitance and local weight) The arguments put forward in the 'hyperbolic interlude' seem weak. There are many many ways to compare to graphs, and even if we accept that the relevant way is to compare the effect of linear operators it seems unclear how the authors end up with their particular distance (there also other ways to compare the angular distance between vectors). As the authors use exclusively this distance to compare their results to other methods this seems a bit like cherry-picking, in particular, since the other methods were not developed with this error metric in mind (and thus a claim that their method is superior based on this metric alone seems strange). I would suggest the authors show at least the comparison in the original error metric with the quadratic form. It would also be good it a comparison to more than one sparsification technique would be shown -- there are many more methods out there; and in particular those combined sparsification / coarsening techniques the authors cite themselves seem relevant for comparisions but are omitted. Finally it would be beneficial if instead of just using some empirical graphs, the authors could use some synthetic graph construcitons in which they have a finer control about the properties of the graph to be sparsifieda / coarsened, such that we can see what are the relevant effects here. For instance the coarsening should work very well if the graphs have 'community structure' etc. Optional suggestion: Some more discussion about the relationships of coarsening to community detection and formal model order reduction of LTI systems could also enhance the manuscript. Clarity. ======== The paper is written quite clearly, overall. I have several issues though: The cost function (7) is meant to be optimized over what exactly? a single edge? In general I found the explanations in section 3.4 could have been more detailed. In the section section 3.5., the authors state that one has to chose the 'appropriate pseudoinverse' but then do not say why their choice is appropriate. For the Sherman-Woodbury-Morrison for rank-deficient matrix -- please refer to the following paper: Meyer, Jr, Carl D. "Generalized inversion of modified matrices." SIAM Journal on Applied Mathematics 24.3 (1973): 315-323. Originality. ============ The idea underpinning the paper is original, even though I feel the cost function the authors propose seems to be somewhat ad-hoc. Significance ============ This is clearly a significant topic, in my opinion, as many graph based algorithms could be augmented by such a technique. Conclusion ========== A good paper with some shortcomings that need to be adressed. Otherwise I think this is an interesting contribution. [update post discussion] After considering the authors' response and the discussion, I believe this could be an interesting contribution, provided the authors improve the paper as indicated in their response letter; in particular with respect to the clarity of the presentation.

Reviewer 3



- The authors present an algorithm (based on an analysis of the pseudoinverse of a graph after randomly perturbing edges in the graph) that can simultaneously be used for graph sparsification and graph coarsening. - Is it clear that $w_e\Omega_e$ will not equal 1 to ensure finiteness in (4)? - I am having trouble following the claim made in Eq. (6). The $M_e$ are fixed matrices with $M_e=w_e (l_i-l_j)(l_i-l_j)^T$ where $l_i$ is the $i$-th column of $L^{\dagger}_G$ and $e=(i,j)$. The randomness lies with the $f$ distributions (which can be assumed uncorrelated) or are you saying it is reasonable to assume a generative process on $G$ that would lead to uncorrelated $M_e$? - In Figure 1, $\beta_{1d}=\emptyset$ for nodes. How are we to interpret the split in the intermediate range in this case? - Can you provide details (either a citation or a derivation in the appendix) of the claim that $d_h(L_G \vec x, L_{\tilde{G}}\vec x)\leq\log(1+\epsilon)$ implies that $\tilde{G}$ is an $\epsilon$-approximation of $G$. This is a key property of the proposed hyperbolic metric, as it ties the classical work to the present experimental comparisons. - When you introduce the problem of graph coarsening, you list a bevy of citations from the literature [9, 10, 3, 11, 12, 13, 14], yet your experiments for graph coarsening only compare against [47] (a paper from 1998) and [24] (an unpublished arXiv preprint). It is hard to gauge if these are sufficient; I understand that more comparisons are included in the appendix (to [24] again and to another unpublished arXiv preprint). Comparing against more algorithms would be helpful in gauging the effectiveness of your approach. - Similarly, for graph sparsification you compare your approach against [20], an algorithm from 2011. Comparisons against more (and more recent) approaches would be helpful in gauging the effectiveness of your approach. Figure 3 in the Appendix (referenced as evidence that sparsification preserves the spectrum) shows that coarsening preserves the spectrum. - It is not clear why Algorithm 2, as presented, avoids the pitfalls of Algorithm 1 (graph becomes disconnected, contraction issues)? Your appendix section A.2 seems to suggest that a non-uniform sampling of $s$ (an independent edge set) is sufficient to ensure no disconnectedness/pitfalls, yet in your main algorithm you have $s$ being selected uniformly-at-random? - Small typos: Equation (11): I believe there is a "-" missing in the numerator of the third term.

[Author Response · NeurIPS 2019]

We thank all three reviewers for your thorough and thoughtful comments. We have already incorporated them into our
revised version of the paper, and are very appreciative of the improvements. Due to space, we group concerns together.
**1. Section 3 R1,R2,R3**: We significantly restructured this section to clarify the following issues.
**1.1. Simultaneous reduction of edges and nodes R1**: When comparing with sparsification, we restrict our method to
only deletion/reweight. Otherwise, we allow both deletion and contraction, reducing both edges and nodes. We now
emphasize that we simply *prioritize* reduction of edges or nodes (as opposed to reducing only one or the other).
**1.2. Section 3.1** (now 3.2) **R3**: Indeed, deleting an edge with $w_e\Omega_e = 1$ would disconnect the graph and invalidate the
use of the S-M-W formula, but this is precluded by the requirement that $\mathbb{E}[f] = 0$. We added this to the text.
**1.3.** $\mathbb{E}[\|\sum \boldsymbol{\Delta L}^\dagger\|_{\mathbf{F}}^2] \approx \sum \mathbb{E}[\|\boldsymbol{\Delta L}^\dagger\|_{\mathbf{F}}^2]$ **approximation R3**: The randomness lies in probabilistic choice of edges.
**R3,R2**: Our "reasonable assumption" comment was indeed imprecise. We removed it, and added an study using
ER and real-world networks, showing that our approximation is either nearly exact or a conservative estimate.
**1.4. Section 3.4 and choice of cost function R1,R2**: This choice of cost function naturally arises when minimizing
the expected squared error for a given expected number of reductions. **R2**: Moreover, as the squared Frobenius error
empirically sums, minimization for each edge acted upon can be seen as a probabilistic greedy algorithm for minimizing
the cost function of the final reduced graph. **R1,R2,R3**: A detailed derivation of the single-edge minimization is now in
the appendix. In the main text, we better explain the regimes for $\beta$. We also fixed the sign error, thank you so much, **R3**!
**1.5. Why unbiased (ie, $\mathbb{E}[\boldsymbol{\Delta L}^\dagger] = \boldsymbol{0}$)? R1**: This condition is necessary for the answers 1.2–1.4. Additionally, the desire
of an unbiased algorithm is quite standard, eg, Spielman et al. reweight the edges so as to ensure $\mathbb{E}[\boldsymbol{\Delta L}] = \boldsymbol{0}$.
**1.6. Section 3.5 R1,R2**: This choice ensures that contraction yields a node-weighted $\boldsymbol{L}^\dagger$ consistent with the $w_e \to \infty$
limit. We now focus the section on the definitions needed for our algorithm, also including how to convert the coarsened
$\boldsymbol{L}$ to a matrix of the original size. In the appendix, we give a detailed rationale for them and explicit numerical examples.
**2. Algorithm 2 and solving issues with multiple simultaneous contractions and deletions R1,R2,R3**: We removed
Algorithm 1 (as it was not used), and incorporated a modification (previously in the appendix) to Algorithm 2 that
ameliorates these issues. In particular, we change our edge sampling method from uniform to a random independent
edge set. This explicitly solves the issue with multiple contractions, and empirically solves the problem of multiple
deletions. **R1**: We also refined our experimental studies of the hyperparameters and included a detailed discussion.
**3. Algorithmic complexity and approximating $\boldsymbol{L}^\dagger$ R2**: The S-M-W formula is needed to derive the optimal probabilis-
tic actions to an edge, but it is actually not necessary for the algorithm itself; the more scalable implementation directly
constructs approximations to $\Omega_e$ and $m_e$ at each iteration, taking $\widetilde{\mathcal{O}}(\langle k\rangle|E|)$ time. We add more in-depth analysis and a
study suggesting that such an approximation does not notably decrease the quality of reduction (appendix). As stated,
our framework cannot start with the empty graph (however, even algorithms that start with an empty graph often need
to store the full graph at some point). We now have a discussion section where we clarify these points.
**4. Frobenius norm, error bounds, and preferential preservation of global eigenvalues R1**: Using a perturbative
analysis, we now bound the change in the eigenvalues in terms of our cost function. We also show that the bound on the
relative error of the global (top) eigenvalues of $\boldsymbol{L}^\dagger$ is smaller, hence preferentially preserving the portion of the spectrum
associated with large-scale structure.
**5. Tradeoff between error and reduction and when to stop R1**: As mentioned in answer 1.3, our approximation
holds well. As it is trivial to keep a running total of this error estimate, one may simply terminate the algorithm when
the Frobenius error reaches the desired limit. We included this in our discussion of Algorithm 2 (now 1).
**6. Our proposed metric and its relationship with standard measures R3**: We slightly modified the definition of our
metric (giving additional desirable properties, eg, triangle inequality). We added a proof of its relationship with the
standard notion of spectral approximation (appendix). **R2**: We also reproduce our figures using standard measures.
**7. More comparisons with other methods and synthetic graphs R2,R3**: We added to all of our coarsening compar-
isons a recent method by Loukas, which is similar in spirit to our approach, and show that our method more accurately
preserves the global structure. **R3**: Most of these citations were about *using* coarsening for other tasks, as opposed to
proposing new coarsening schemes. **R2,R3**: Indeed, it would be good to compare with several sparsification methods.
However, in our literature search, we found that they either do not explicitly preserve properties associated with the
Laplacian (and hence, would be an unfair comparison), or are very similar to Spielman's algorithm, with the goal of
computational efficiency (at the expense of some accuracy). As speed is not the primary focus of our paper, we compared
with the exact implementation. We added this discussion to the text. We are working on a comparison with a method
that sequentially combines sparsification and coarsening (the closest we found to our unifying approach). **R2**: and on
experiments comparing our method with other coarsening methods using SBMs. All these will be included if our paper
is accepted! For either sparsification or coarsening, if you know about methods that we are unaware of, we would be
happy to add them to our comparisons. **R1,R2,R3**: Please zoom for some new figures:

55

[Meta-Review · NeurIPS 2019]

Graph sparsification has a well developed literature with strong theoretical guarantees. In contrast, the literature on graph reductions (coarsening) is much more ad-hoc, even though from a practical point of view coarsening could be very useful. The present paper introduces an imaginative way to combine these two tasks. While there are no theoretical guarantees (which is what Reviewer 1 complains about) and some of the details like the computational complexity or a detailed explanation of the rational behind the loss function/algorithmic details are not fully fleshed out, there are plenty of interesting mathematical insights here that are worth sharing with the community. I am also a bit mystified by the reason for the hyperbolic distance that the authors introduce and wonder if one could construct a bottom up version of the algorithm, because that would be very important for large problems. Overall, I think this is a valuable contribution because it might help the field of graph coarsening get "unstuck".